# Metabolt: An In-Situ Instrument to Characterize the Metabolic Activity of Microbial Soil Ecosystems Using Electrochemical and Gaseous Signatures

**DOI:** 10.3390/s20164479

**Published:** 2020-08-11

**Authors:** Miracle Israel Nazarious, María-Paz Zorzano, Javier Martín-Torres

**Affiliations:** 1Group of Atmospheric Science, Department of Computer Science, Electrical and Space Engineering, Luleå University of Technology, 97187 Luleå, Sweden; maria-paz.zorzano.mier@ltu.se (M.-P.Z.); javier.martin-torres@ltu.se (J.M.-T.); 2Centro de Astrobiología (CSIC-INTA), Torrejon de Ardoz, 28850 Madrid, Spain; 3School of Geosciences, University of Aberdeen, Meston Building, King’s College, Aberdeen AB24 3UE, UK; 4Instituto Andaluz de Ciencias de la Tierra (CSIC-UGR), 18100 Granada, Spain

**Keywords:** Metabolt, space, electrical conductivity, redox potential, gas monitoring, microbial metabolism, astrobiology, greenhouses, planetary analogue research, planetary exploration

## Abstract

Metabolt is a portable soil incubator to characterize the metabolic activity of microbial ecosystems in soils. It measures the electrical conductivity, the redox potential, and the concentration of certain metabolism-related gases in the headspace just above a given sample of regolith. In its current design, the overall weight of Metabolt, including the soils (250 g), is 1.9 kg with a maximum power consumption of 1.5 W. Metabolt has been designed to monitor the activity of the soil microbiome for Earth and space applications. In particular, it can be used to monitor the health of soils, the atmospheric-regolith fixation, and release of gaseous species such as N_2_, H_2_O, CO_2_, O_2_, N_2_O, NH_3_, etc., that affect the Earth climate and atmospheric chemistry. It may be used to detect and monitor life signatures in soils, treated or untreated, as well as in controlled environments like greenhouse facilities in space, laboratory research environments like anaerobic chambers, or simulating facilities with different atmospheres and pressures. To illustrate its operation, we tested the instrument with sub-arctic soil samples at Earth environmental conditions under three different conditions: (i) no treatment (unperturbed); (ii) sterilized soil: after heating at 125 °C for 35.4 h (thermal stress); (iii) stressed soil: after adding 25% CaCl_2_ brine (osmotic stress); with and without addition of 0.5% glucose solution (for control). All the samples showed some distinguishable metabolic response, however there was a time delay on its appearance which depends on the treatment applied to the samples: 80 h for thermal stress without glucose, 59 h with glucose; 36 h for osmotic stress with glucose and no significant reactivation in the pure water case. This instrument shows that, over time, there is a clear observable footprint of the electrochemical signatures in the redox profile which is complementary to the gaseous footprint of the metabolic activity through respiration.

## 1. Introduction

The investigation of the state of activity of the natural microbial ecological systems that inhabit the soils is of interest to explore a vast amount of unknown ecosystem relationships and adaptive responses that are presently active on Earth [1]. These relationships can be further characterized subjecting this system to different environmental conditions (nutrients, temperature, osmotic pressure, dry or humid environment, pH, etc.) [1], investigating the limits of these environmental factors where life can thrive and the mechanisms that are activated then. This subject is at the core of extremophile research on planetary analogues [2,3,4,5] and is particularly relevant for astrobiology to define the habitability limits on other planetary environments [6,7].

Of particular interest to our research is the detection of living cells within the soils. The categorization of microorganisms in soil that has been widely employed is the division between the major decomposer groups: fungi and bacteria [8,9]. While these two groups are dominant in different soils, the fungal-to-bacterial ratio ranging from 1.0 to 2.3 or much higher acts as a good indicator of environmental changes in the soil [10,11]. Despite significant improvement in microbiology and molecular laboratory practices, only ≈1% of existing bacterial strains can be cultivated in the laboratory with standard broth and agar [12,13,14,15]. Additionally, some microorganisms have adapted to specific growing conditions, such as slow growth in a low resource environment, or anaerobic conditions, and thus they cannot be cultured with standard high-nutrient broth assays, where other faster and aerobic growing organisms dominate the micro-cosmos [1,16]. Most, if not all, microorganisms in soil are dormant [17] at a given point in time and thus, any ideal system to monitor the microbiome of soils should allow for all the natural phases to take place spontaneously, monitoring the system with minimal intervention. The identification of the environmental factors that activate some metabolic processes is, however, limited by the current tools and laboratory practices. This fact impedes detection and characterization of a wide variety of species as well as the natural microbial activity, tuned in response to changing environments depending on the ecosystem where they live and the physical signals from neighboring microbes [18]. We present the Metabolt as a portable incubator to characterize the metabolic activity of microbial ecosystems on the regolith using electrical conductivity, redox potential, and gaseous concentration measurements.

Soil microbial activity has been measured using a variety of sensors and methods so far. The electrical conductivity (EC) and redox potential (Eh) have been demonstrated to be viable measurement techniques to monitor the microbial metabolic activity. Back in the 1970s, it was demonstrated that cellular metabolism results in an increase in EC and water-soluble Ca and Mg in the supernatant liquid (80 mL solution with 0.5% glucose) above 30 cm^3^ of natural soil in the dark at 30 °C without agitation [19]. This method was thus proposed to detect life in extraterrestrial soils [19]. In more recent times, other studies have targeted the investigation of soil biogeochemistry and transient redox conditions caused by the water table fluctuations [20] and observed a decreasing soil Eh after the addition of glucose [21]. Enright et al. [22,23] demonstrated a distinguishable biological signal by immersing a pair of electrodes in an iron-oxidizing biofilm. Graphite electrode experiments have also been used to investigate the rates of anaerobic microbial activity in a diversity of anoxic sediments [24]. Soil respiration serves as an accurate indicator to study soil microbial ecology and soil health. There are systems in place for continuous in-situ monitoring of CO_2_ fluxes autonomously [25,26,27]. Reiser et al. [28] combined the Eh measurements with the O_2_ diffusion rate. Biogenic volatile organic compounds (VOCs) and gases released in soil are known to be linked to microbial activity that can be sensed with electronic nose (E-nose) technology [29]. Another existing method extracted components of electron transport chains for remote detection of chemical signatures of life in soil. Techniques such as chromatography or electrophoresis to separate extracted compounds, with final detection by voltammetry and tandem mass-spectrometry were proposed [30]. Recently, nanomaterial incorporation into biosensors has been proposed to enhance the performance of biosensors due to their unique physicochemical properties [31]. Vision methods that are used for non-contact monitoring of materials and structures could also be supplementary to Metabolt’s future application in space greenhouses. The use of small sized cameras and advanced algorithms could provide a possibility to monitor the growth of biofilms, fungi structures, and microbial mats or other macroscopic structures on the surface of the soils [32,33]. The design of Metabolt takes a novel step forward combining some of these different approaches under one system to comprehensively analyze multiple physico-chemical parameters such as electrical conductivity and redox potential along with gas fluxes. We also included H_2_O measurements because with regolith samples containing salts such as on Mars, the relative humidity is important to assess their hydrate or brine or dehydrated states.

One of the most direct applications of Metabolt is the detection of signatures of living microbial cells within soils, which is particularly critical for space exploration and in-situ investigation in Martian analogues and, in general, in remote regions on Earth. For instance, recent studies of in-situ metabolic activity performed in the driest parts of the Atacama Desert demonstrated that active metabolism can still occur in specialized microhabitats, such as the interior of salt nodules [34]. Similar efforts have been made to deploy portable microbiological instrumentation in the Canadian Arctic permafrost [35]. The role of microbial metabolism in the changing climate of the Arctic and its contribution to an increased emission of greenhouse gases is still unknown. This type of instrument is thus also of interest for long-term in-situ monitoring of the emission of gases by microbes in the Arctic region, in a natural environment. These examples show the need for robust, multi-purpose, portable, autonomous instruments such as Metabolt to monitor the microbial activity and the gaseous interchange with the atmosphere in its natural environment with minimal intervention. Finally, it is also interesting to characterize the response, of a natural microbiome community or a specific incubated strain, subjected to a particular treatment. We will show in this article the construction of Metabolt and will illustrate its operation with one specific set of experiments.

We designed Metabolt with the following functional requirements. The instrument shall:mimic natural growth conditions;register information associated with time variability;allow investigating spatial differentiation and different experimental conditions;monitor in a non-invasive, non-specific, and large-scale way, the metabolic activity of a system; andmonitor the gaseous interchange with the atmosphere.

Metabolt allows monitoring over time, the electric conductivity and the redox potential of a soil sample, in order to investigate the different stages of metabolic activity of a microbial community. Additionally, the instrument monitors atmospheric composition in the headspace above the soil, which allows investigating gaseous interchange between the atmosphere and soil, and to confirm the footprint of metabolic activity through respiration. The design of Metabolt is flexible and can be adapted to other specific investigations with different gas sensors depending on the target of interest. On its present form, Metabolt allows for the simultaneous comparison of two experiments in independent containers. Finally, the instrument was designed to be compact, modular, autonomous, and with small volume, mass, and power requirements as it is usually needed for space exploration and field-site campaigns.

## 2. Materials and Methods

### 2.1. The Instrument

Metabolt consists of two units: an incubator with two experiment containers, and an electronic box which houses the signal processing circuitry, central commanding and processing, and data management system (Figure 1). The battery power backup system and an in-built flash memory facilitate the continuous operation of the experiment in the event of a power outage, making the instrument reliable and ultra-portable to any field test sites. The two electrochemical measurements of the soil are performed with custom-built sensors: 99% pure copper sheet electrodes and platinum electrodes for the electrical conductivity and the redox potential measurements, respectively. Since the circuits for measuring the electrical conductivity and the redox potential could generate their local electromagnetic fields, there is a possibility of interference which is avoided using a sensor shield from Whitebox Labs [36]. This provides electrical isolation between the sensors and removes the external electrical noise that can interfere with the readings. The current design of Metabolt contains O_2_ and CO_2_ gas sensors. However, depending on the target of interest, other gas sensors such as NH_3_, N_2_O, CH_4_, etc. can be integrated to the system directly. Depending on the number of sensors used, the lid of Metabolt that is accommodating the sensors may get larger and consequently the experiment container too. The technical specifications of the sensors are summarized in Table 1, and the physical dimensions of the instrument are provided in the Appendix A. The mass of each of the experiment containers with lid and sensors is 250 g, the casing and connectors weigh 400 g and the electronic box, 500 g. Since the dimensions of the lid was driven by the number of sensors used, and the dimensions of the experiment container was driven by the size of the electrodes to have a good enough cell constant to measure electrical conductivity in a valid range that is typical for moist soil samples, the allowed weight of soil samples and solutions in each container is fixed to be 250 g. These make the overall weight of Metabolt 1.9 kg and its maximum power consumption 1.5 W. The operation of the instrument has been demonstrated in the international astrobiology field campaign MINAR 5 [37] organized by the UK Centre for Astrobiology, in the Boulby salt mine (UK), at 1.1 km depth.

### 2.2. Sensor Calibration Procedure

The soil electrical conductivity probe was calibrated with an EZO™ conductivity circuit [38] for the known geometrical cell constant (from the probe dimensions, K = 9 cm/(5 × 3 cm) = 0.6 cm − 1) and with the conductivity standards: 1413 µS/cm (HI7031, Manufactured by: Hanna Instruments, Rhode Island, US; Sourced from: HannaNorden AB, Kungsbacka, Sweden) and 5000 µS/cm (HI7039, Manufactured by: Hanna Instruments, Rhode Island, US; Sourced from: HannaNorden AB, Kungsbacka, Sweden) for a two-point calibration at 25 °C. The temperatures of the standard solutions were simultaneously recorded with a calibrated temperature sensor (DS18B20, Maxim Integrated; Manufactured by: SparkFun Electronics, Colorado, US; Sourced from: Digi-Key Sweden), and the conductivity measurements were temperature-compensated in the circuit. The redox and reference probes were calibrated with an EZO™ ORP circuit [39] against the Oxidation Reduction Potential (ORP) standard solution, 200–275 mV (HI7020, Manufactured by: Hanna Instruments, Rhode Island, US; Sourced from: HannaNorden AB, Kungsbacka, Sweden) at 25 °C. The Eh (redox potential w.r.t. standard hydrogen reference electrode) was derived from ORP + 210 mV [40,41,42,43] (w.r.t Ag|AgCl wire with 3M KCl reference electrode).

### 2.3. Sample Preparation

For a demonstration of the operability of the instrument, we present here a set of laboratory experiments where we used natural Arctic soil with three different treatments to illustrate the forced and naturally observable changes in the metabolic activity of soils and the gaseous interchange with the atmosphere. The three different soil samples are (I) unperturbed sub-arctic soil collected in Luleå (65°37′03.72″ N, 22°08′25.81″ E), Sweden. The unsaturated silty sand sampled at a depth of 0.05–0.15 m had the following properties: organic matter content: 2.1 ± 0.01%, moisture content: <10.2%, and pH = 5.7 ± 0.1; (II) soil collected at the same location of (I) after heat sterilization (under uncontrolled relative humidity conditions), following the standards of the European Space Agency (ESA) for the bioburden reduction (ECSS-Q-ST-70-57C) [44]. The soil was heated in a climate chamber (Heraeus HT 4010) at 125 °C (incl. temperature distribution uncertainties) for 35.4 h. This procedure is named the Dry Heat Microbial Reduction (DHMR), and it is used to reduce the bioburden of hardware, components, or products which are sent to space [44]. For this specific condition, the number of bacterial spores are expected to be reduced by 4–6 orders of magnitude. The experiment was not run in cleanroom conditions, the instrument itself was reused from one test to another; therefore, the recontamination of the samples or survival of spores cannot be discarded; (III) soil collected at the same location of (I) after adding 40 mL of 25% CaCl_2_ brine, which in pure brine state at laboratory temperatures should have a water activity of aw = 0.8 [45]. Our measurements suggest a resulting water activity in the soil of aw~0.85 (relative humidity of about 85%) (see the Appendix A). The purpose of this experiment was to illustrate that water activity is one of the primary physicochemical factors that set a hard limit to life [1,46,47]. The scheme of the set of tests performed to increase the statistical nature of this demonstration study is shown in Table 2. For each different soil treatment, we ran three experiments, one with soil sampled in August 2017 (E1) and two technical replicates (E2 and E3) with soil sampled in Summer 2018. For simplicity, since all the replicates showed similar responses, we skipped showing one experiment each under every treatment. As will be shown later, the three experiments under different conditions/treatment showed reproducibility in their responses.

### 2.4. Experimental Setup

We divided the treated samples into two parts, to fill a volume of 180 cm^3^ (about 200 g in dry weight) in each experiment container. The depth of the soil sample was about 32 mm. Before the experiment, the soil samples were sieved to remove particles of diameter >5 mm. The whole experimental setup was placed in an empty room with low thermal and gaseous diurnal variations and the experiment was initiated at ambient conditions. Each sample was incubated independently after the addition of 40 mL of deionized water (processed by Silhorko—Eurowater A/S; Type: Silex 1B), and the response was monitored for at least 5 days.

For every experiment, in one of the two experiment containers, glucose was added as an organic carbon source; diluted in the 40 mL deionized water with 0.5% concentration by volume. The general chemical equation for the metabolism of glucose by heterotrophs can be written as:(1)$$C_{6}H_{12}O_{6}+6O_{2}\;\to 6{\mathrm{CO}}_{2}+6H_{2}O+2870\;\mathrm{kJ}$$

Thus, we expect to see anti-correlation signatures in oxygen, O_2_, and carbon dioxide, CO_2_, if this metabolic pathway is activated. We demonstrate the working principle with these two gases. However, similar studies can be performed for anaerobic organisms, monitoring: nitrogen oxides, methane, ammonia, and other volatile products which can contribute to short and long-term changes in the composition, chemistry, and radiative balance of the atmosphere of the Earth.

### 2.5. Measurements, Data Sampling, and Data Treatment

In addition to the two electrochemical measurements, six environmental parameters are measured simultaneously: air and soil temperature (T); pressure (P) and relative humidity (RH) in the headspace; and concentrations of oxygen (O_2_) and carbon dioxide (CO_2_) respiration products in the headspace.

Due to the continuous flow of an alternating current through the electrodes for the EC measurements, the surface of the copper sheet electrodes is prone to corrosion. The system software was optimized to run the experiment for 10 consecutive measurements and hibernate for an hour (to limit the power consumption and minimize the electric current exposure) before continuing with the measurements.

Low-cost commercial gas sensors are prone to offset degradation after a longtime saturation. The offset of the used gas sensors is determined by comparing with a new gas sensor, at ambient laboratory conditions. This offset is corrected later on the resulting data assuming a linear degradation. The gas sensors may also show a transient response at the beginning of its operation from the idle state until some initial equilibrium is reached. The first changes for about 11 h caused by the water redistribution had to be discarded for proper interpretation. The first few minutes of every hour produces 10 data points. In some experiments, we observed a sizeable varying range of values within an hourly dataset.

For every experiment, all the data points were used directly, and the trend is shown with a smoothed cubic Bezier curve in Figure 2, Figure 3 and Figure 4.

## 3. Results

We summarize some of the main observations of the three study cases.

### 3.1. Gaseous Signatures as a Footprint of Metabolic Activity through Respiration

Figure 2 shows the measurements produced during incubation of 9–15 days of the two different repetitions of experiment I (“Unperturbed soil”), after the addition of pure water (control) or water with carbohydrate additives (glucose). On the very first day, after the addition of water, there is a rapid activation of the metabolism which can be detected by a sudden, exponential increase of the atmospheric CO_2_ level from an hourly average of roughly 550 ppm in ambient environment (see Appendix A) to about 8000 ppm (Figure 2d,h). The CO_2_ values hit a ceiling around 8000 ppm (instead of 10,000 ppm maximum range of the sensor) because of its initial offset. The data shows that, since the chambers are not airtight in this prototype, there is an apparent exchange of both gases with the external environment. If the system was fully sealed, the headspace air would likely run out of oxygen with glucose addition and go anaerobic. It seems that the CO_2_ also reaches a kind of equilibrium level at ca. 8000 ppm where production equals leakage from the chamber. There is a diurnal variation in the gas signal data—clearer in O_2_ than in CO_2_. While external ambient O_2_ will vary due to light photosynthesis/dark respiration, the data suggests a more direct response of the soil to temperature. This is very obvious in E3 for O_2_ and to a lesser extent for CO_2_. The observed CO_2_ increase has the shape of typical aerobic growth curves and may thus be an indicator of the intense metabolism caused by cell replication [19,48,49]. This concentration is diluted over time as the enclosure, of this prototype version of Metabolt, is not airtight. Over the days, as the metabolic activity is reduced, the CO_2_ level decreases to an average value of about 3000 ppm with small diurnal modulations. In parallel to this first CO_2_ change there is a significant reduction of the O_2_ level (Figure 2c,g), which is particularly detectable for the cases where glucose has been added. The diurnal modulation of the O_2_ concentration is evident in this case.

### 3.2. Electrochemical Signatures as a Potential Indicator of Microbial Metabolism

In parallel to this respiration signature, we also observe the changes in the electrical conductivity (Figure 2a,e) and the redox potential (Figure 2b,f), with sustained changes over the days even beyond 2 weeks. On some days, the electrical conductivity can change up to 20 µS/cm between the day and the night activity (Figure 2e). Since electrical conductivity measurements are temperature sensitive, some of these variations may be attributed to the change in temperature between the day and night. The behavior in electrical conductivity is also accompanied by an anti-correlated 0.5% change in the O_2_ concentration. The diurnal variability of Eh can be as significant as 100 mV for some cases, but it seems less periodic and generally shows some variability around an average value. Eh is generally higher in the case of control as compared to the glucose case. However, in some experiments, the order was reverse. The anomalies in the initial values of EC and Eh are usually due to the way the soil sample is in contact with the electrodes. Since the contact of the soil with the electrodes for EC and the probes for Eh depends on the distribution of moisture within the soil, the contact is not always the same. In the figures we present the true value of EC and Eh as measured but the change in measurement over time might be more relevant for interpretation. The diurnal variabilities suggest that daily changes in temperature plays a significant role in the response, However, the sustained changes over time, with modulations of the EC and the Eh values can be used as an indicator of the microbial activity within the regolith as is explained in further sections.

### 3.3. The Difference in Response Due to Temperature (Unperturbed vs. Thermal Stress)

Sterilized soils are expected to contain many labile nutrients (which is in one experiment enhanced with the glucose). These will enable rapid colonization by contaminants (or survivors, if any). In such a system, the expected behavior at ambient temperatures, is to have 2–3 days delay, before the growth actually becomes measurable. This is indeed observed in the experiment. Figure 3 shows the measurements produced during an incubation of 10–20 days of the two different repetitions of experiment II (“Thermally stressed soil”), after the addition of pure water (control) or water with carbon hydrate additives (glucose). Let us point out that after the applied DHMR process we expect a reduction of 4–6 orders of magnitude in the number of bacterial spores in the soil. This has a definite impact on the CO_2_ signal, indeed. However, since the system is non-sterile the recontamination is possible with an initial delay. Comparing the metabolic signatures of samples under thermal stress treatment with the unperturbed revealed that the exponential growth phase was delayed by up to 59 h in the glucose experiment and 80 h in control (Figure 3c,d). The reactivation of metabolic activity was sped up in both experiments by the addition of glucose, indicating the significant role of the nutrient availability on the cell reproduction process. Once the number of cells has increased significantly, we observe a saturated value of the CO_2_, which again dilutes over time. Additionally, now and in parallel to this initial (but lagged) CO_2_ release there is a significant reduction of the O_2_ concentration which is particularly detectable for experiment 1 (Figure 3c), where all the data measurements are available. Here, there is also a time difference of 2 days in the activity between the experiments: the case with glucose reaches the minimum earlier. In the thermally stressed test, there is a sustained variability of the EC and the Eh, although there is no clear diurnal variation as opposed to unperturbed soil where the diurnal variations in EC and Eh due to the effect of daily temperature changes were more evident. This may suggest that the overall response of the system consists of the true microbial response and the diurnal variations due to temperature changes accentuates it. Temperature-controlled experiments are required to isolate the temperature effects in order to observe the true microbial response. Interestingly for some experiments, the EC, O_2_, and CO_2_ values converged towards common values around the end of the experiment indicating that glucose was consumed and both ecosystems behave similarly.

This example is particularly interesting to illustrate how the ecosystem changes when stress conditions are applied. After the exposure to stress, microbes may be in a senescent state where they remain partially active to maintain viability and protect against stress conditions. Bacteria have numerous strategies that are activated to cope with the stressed conditions including: the formation of cysts and spores, changes in cellular membranes, expression of repair enzymes for damage, and synthesis of molecules for relieving stresses [50]. Such metabolism still necessitates appropriate amounts of energy [51], which was observed from the highly variable Eh in Figure 3b,f (red curves).

### 3.4. The Difference in Response Due to Reduced Water Activity (Unperturbed vs. Osmotic Stress)

Figure 4 shows the measurements produced during incubation of 11–16 days of the two different repetitions of experiment III (“Osmotically stressed soil”), after the addition of pure water (control) or water with carbon hydrate additives (glucose). Let us point out that after the applied stress, different samples show different responses depending on the dominant adapted species. For instance, Figure 4(E1) managed to show a robust aerobic response after 4 days, whereas Figure 4(E3) did not show such response. In fact, in E1 the CO_2_ release and the decrease in the O_2_ was much more pronounced for the case where additional nutrients were available in the form of glucose. Clearly, the reactivation seems harder in this case as compared to the DHMR case (where the viable cells were reduced by 4–6 orders of magnitude). Let us point out that whereas on experiment II (“Thermally stress soil”) the stress disappears, in this case, the brine is always within the soil producing continuous, unavoidable stress to the existing microbiome. The growth process in brines can be divided into three phases [52] based on the microbial response to osmotic stress: (1) initial reaction of either efflux (hyperosmotic stress) or influx (hypo-osmotic stress) of cell water along the osmotic gradient, leading to rapid shrinkage or swelling of the cytoplasm; (2) biochemical readjustments occur to restore the turgor and volume. Here we consider hyperosmotic conditions resulting in an increased transport or synthesis of the compatible solutes; (3) growth is resumed under new conditions maintaining the required biochemical adjustments in response to the environment 1. These phases can be observed in Figure 4a,e which shows, in detail, the electrical conductivity pattern and the modulations overwritten on the increasing electrical conductivity curve. In E3, there is a very rapid initial release of the CO_2_ and the decrease of the O_2_ followed by the large variations of the EC and the Eh throughout the full duration of the experiment, whereas E1 shows some milder, diurnal modulations in the EC and the Eh. With thermally stressed soil the diurnal variations in EC and Eh were almost not present in the first few days due to the minimum to no microbial activity. However, in osmotically stressed soil with the addition of salt, the ionic activity in the salt may have contributed to the unstable variations in the electrochemical activity observed by EC and Eh. The purpose of this treatment was to observe the response of osmotic stress in soil microorganisms with the application of salt as seen in other planetary environments if any. Though we could not observe a clear trend in the beginning, we could observe a stabilization towards the end that is common to both control and glucose cases. However, other measurements returned valid information showing the effect of osmotic stress with no significant revival in metabolism even after days. We presume that after the sustained exposure to stress, there is a natural selection process in these ecosystems and the response of the survival species is dominant.

## 4. Discussion

This method may be used as a pre-screening protocol to select the samples that should be further analyzed by the more traditional biochemistry, genetic, and molecular biology methods with standard laboratory equipment. For example, it may be used to consider the microbes from soils that have demonstrated activity when exposed to osmotic stress, radioactivity, prolonged exposure to desiccation, the addition of metals, or other products. This approach can also be used in field studies to investigate the metabolic activity of the microbial ecosystems in their natural environment, assisting in the selection of the potential samples or the partially modified conditions. The results described here from the laboratory studies, used a thermodynamically semi-open system of Metabolt, where the vapor and gases can exchange irreversibly with the surrounding and are not supplemented either internally within the soil or from the external atmosphere. This characteristic may alter the measurements of the soil when compared to their natural conditions. Future, portable versions for in-situ monitoring of the natural activity of soils in the field may be designed as an open frame, to hold the electrodes, redox probes, and sensors, with a lid to hold the sensors that monitor the headspace, and with an open bottom to allow for water recharge through the subsurface.

In the laboratory, the methodology implemented in the Metabolt instrument can find applications as a pre-screening procedure or in a complementary fashion with other laboratory techniques:to observe the electrochemical behavior in the samples as a result of microbial metabolism from the instantaneous profile of the EC and the Eh;to observe the global metabolic response to different additives (salts, antibiotic, toxic compounds, metals, etc.) and selective pressures (thermal treatment, pH changes, etc.) which may help to force the dominance of one species or one pathway over other (thermal treatment and osmotic stress were demonstrated in this paper);to understand the time response of the system to external factors like temperature or light, and other inducers of the diurnal variations;to register the response of living cells; to monitor in real time the phases of the lag, exponential growth, and the death of the cellular processes; andto investigate the optimal growth conditions in the solid or liquid media, about the temperature and other additives or parameters (water activity, pH, gases, nutrients, etc.) for the natural samples or selected species.

Some conditions of these applications were demonstrated in the results section of this paper. Our future research will investigate the tolerance of the natural strains to shallow temperature (T), and low water activity (a_w_), which are among the most stringent constraints for microbial cell reproduction on Earth, with the reported limits of T = −20 °C and a_w_ = 0.585 [53,54]. The instrument can also be operated within dedicated experimental facilities with controlled atmospheres and temperatures. For instance, future studies of metabolic activities of the microbiome in the regolith will be investigated by inserting Metabolt within a Martian-environment simulating facility [55,56] and with an Earth pressure, anaerobic chamber. We also plan to run experiments with different regolith to observe variations in response and in temperature-controlled environments within a climate or space environment simulation facility to remove the temperature effects in the measurements to observe the true microbial response.

Due to its portability and autonomy (data are directly logged in the system which can be checked remotely, with no requirement for human intervention after the initiation of the experiment), the Metabolt serves as a good candidate for research in remote regions (like the Arctic, Antarctic, mines, desert areas, etc.) to investigate in-situ, the global activity of wild strains as well as their role on the atmospheric composition in geobiological studies. The concept can be applied to a space qualified model for planetary exploration purposes. Our instrument Metabolt has been recently taken to a field campaign at an analogue for Moon and Mars cave exploration, demonstrating its applicability for fast, in-situ detection of the microbial activity in the ancient salt deposits of a subsurface mine [28]. In October 2017, the international astrobiology field campaign (MINAR 5) explored the Boulby salt mine (UK) in collaboration with the Spaceward Bound (NASA) and the Kalam Centre, India. We investigated in-situ, the viability of cells within a long time preserved saline mineral sample at its natural environment, in the 1.1 km deep salt mine. The Boulby mine hosts the Boulby Underground Laboratory for dark matter studies and an astrobiology laboratory [57]. Metabolt was used to monitor the metabolic activity in-situ in the crushed salt samples (a mixture of halite—NaCl, potash—KCl and polyhalite—K_2_Ca_2_Mg(SO_4_)_4_·2H_2_O) from the mine [37]. These samples were formed in a ≈0.25 Ga-old deep subsurface evaporite deposit and are exposed to the natural radiation emitted from the potash and had no access to visible light or water flow. Upon irrigation with 40 mL of water, and after 4 days of the in-situ incubation, the experiment showed a pattern of the metabolic activity similar to the ones that are described here [37]. The experiment showed a delayed response after 4 days of incubation when the metabolic process in the mine sample is activated enough to be measurable. After activation, the diurnal variation as a function of daily temperature changes could be observed.

The simple approach of the Metabolt instrument has vast applications in the microbial life search on other planets or even for the characterization of the microbiome in the enclosed artificial ecosystems on space environments, such as in the International Space Station (ISS), or exposed to space radiation, such as would happen on the Moon or Mars. Ever since the 1970s Viking lander mission, space agencies have dedicated their efforts to conduct the in-situ experiments on other planets, particularly on Mars—to find the evidence of the present-day biological activity or the life once inhabited on the planet [58]. Changes in the redox potential of the Mars Phoenix mission, Wet Chemistry Laboratory (WCL) Rosy Red sample soil solution were small and transient only at the beginning after the water addition, converging to very stable values as expected for the pure mineral samples with no biological activity [59,60]. This measurement focused on a liquid phase whereas the Metabolt addresses the challenge of monitoring soils directly without any manipulation to the natural environment, and also, allowing to monitor the aerobic and anaerobic metabolic processes with the redox potential measurements. The current semi-open version of Metabolt can be upgraded with sealed containers to be used for experiments in environments with no atmosphere or lower atmospheric pressures such as in the ISS, Moon, Mars, or in simulation chambers.

Other potential applications of the Metabolt could be to characterize the fundamental role of the soil microorganisms on the atmospheric-regolith fixation and the release of gaseous species (such as H_2_O, CO_2_, O_2_, CH_4_, NH_3_, N_2_O, etc.) that affects the climate, atmospheric chemistry, and radiative transfer. It may also be used to quantify the performance of the microbiome in a plantation environment which could be useful for monitoring the greenhouses in the future Lunar and Martian settlements. For that reason, we proposed the Metabolt as a payload for the recent call of the European Space Agency (ESA) for the surface landers of the Lunar Exploration Campaign. Metabolt as a space instrument would need to undergo few modifications from its present form. Among other changes, the experiment container will be made in metal to ensure a proper sealing. Sterilization of the instrument would follow the standard bioburden reduction procedure for flight hardware like DHMR or UV or gamma radiation techniques. As for the soil bacteria, if the instrument is launched to space to a controlled environment like the ISS or a platform on the Moon, the soil can be unsterilized if there is interest to investigate the behavior of terrestrial life forms in space. Indeed, the small size, power, and the mass budget requirements of this instrument together with its robustness, allow the Metabolt to be proposed as a fundamental instrument for microbial life research in space exploration.

## 5. Conclusions

In this work, we described the design, instrument calibration, and the performance of the Metabolt instrument. Metabolt works as an incubator for characterizing in-situ, the metabolic activity of microorganisms. To illustrate its operation, we tested the instrument functionality with a natural ecosystem: Arctic soil (solid) samples, with all the naturally present species. However, similar studies could be performed with pre-selected species in specific growth media or prepared granular solid samples.

In summary, Metabolt characterizes the metabolic activity of a microbial community in a natural growth environment, by monitoring the temporal evolution of the two electrochemical variables: electrical conductivity and redox potential, and the gaseous interchange with the atmosphere. These measurements can be used as a global signature of the microbial activity.

## Figures and Tables

**Figure 1 sensors-20-04479-f001:**
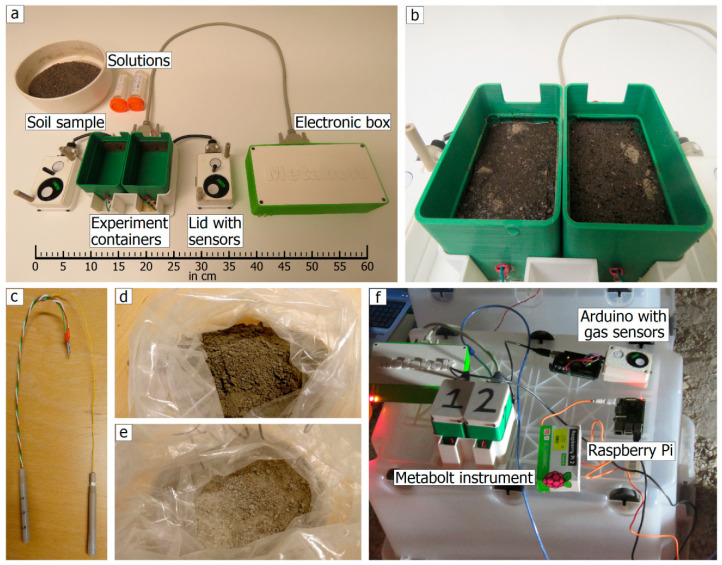
(**a**) Deployed configuration of the Metabolt instrument, with open experiment containers showing the electrical conductivity probes, (in the lid) temperature sensors, redox and reference probes, oxygen and carbon dioxide gas sensors; electronic box; soil sample, water and 0.5% glucose solution. (**b**) Open experiment containers with soil and added solutions ready for the experiment. (**c**) Redox probe with platinum electrodes at three different levels and reference probe. (**d**,**e**) Salt samples (a mixture of halite, potash, and polyhalite) from Boulby salt mine, UK. (**f**) Field-site test at the Boulby salt mine (1.1 km depth), UK as a part of MINAR 5 campaign, showing Metabolt instrument in operation with a Raspberry Pi setup for power and data management.

**Figure 2 sensors-20-04479-f002:**
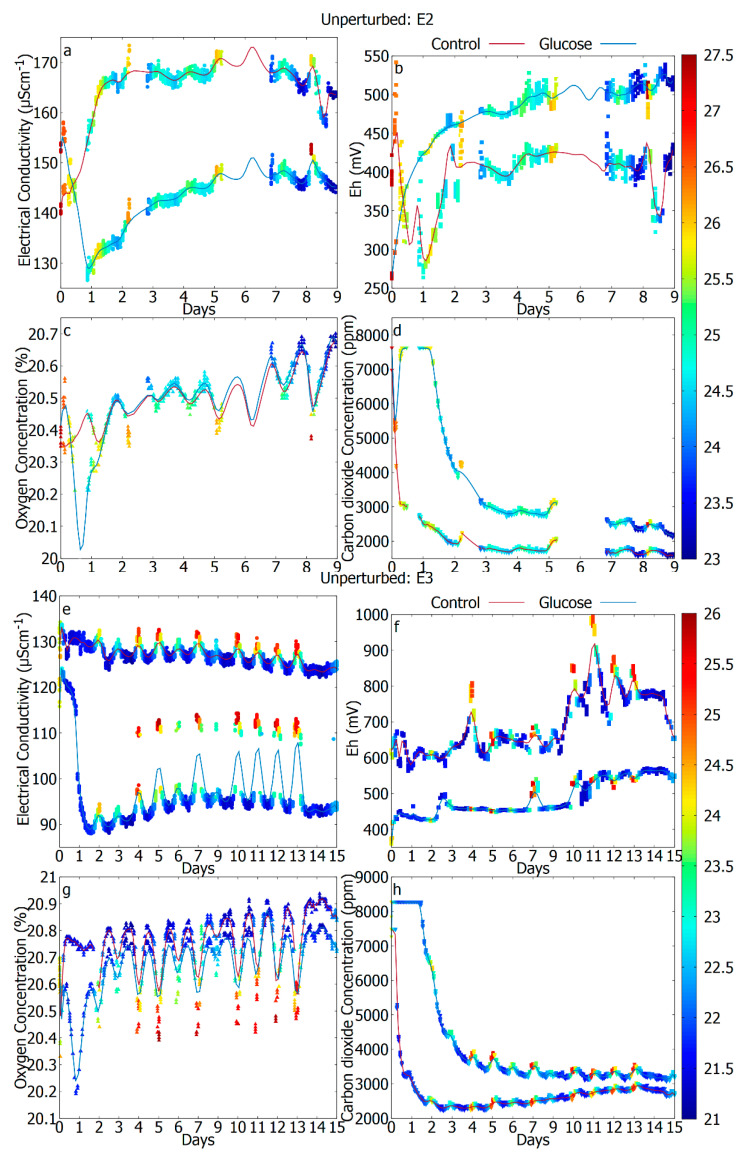
Experiment I. Results showing the different stages of the metabolic activity for control (red) and glucose (blue) cases of the incubated unperturbed soil. E2: (**a**) electrical conductivity (EC), (**b**) redox potential (Eh), (**c**) oxygen concentration, (**d**) carbon dioxide concentration, E3: (**e**) electrical conductivity (EC), (**f**) redox potential (Eh), (**g**) oxygen concentration, (**h**) carbon dioxide concentration. The data gaps in (**d**) were because of the corrupted CO_2_ values. The color bar represents the air temperature (°C).

**Figure 3 sensors-20-04479-f003:**
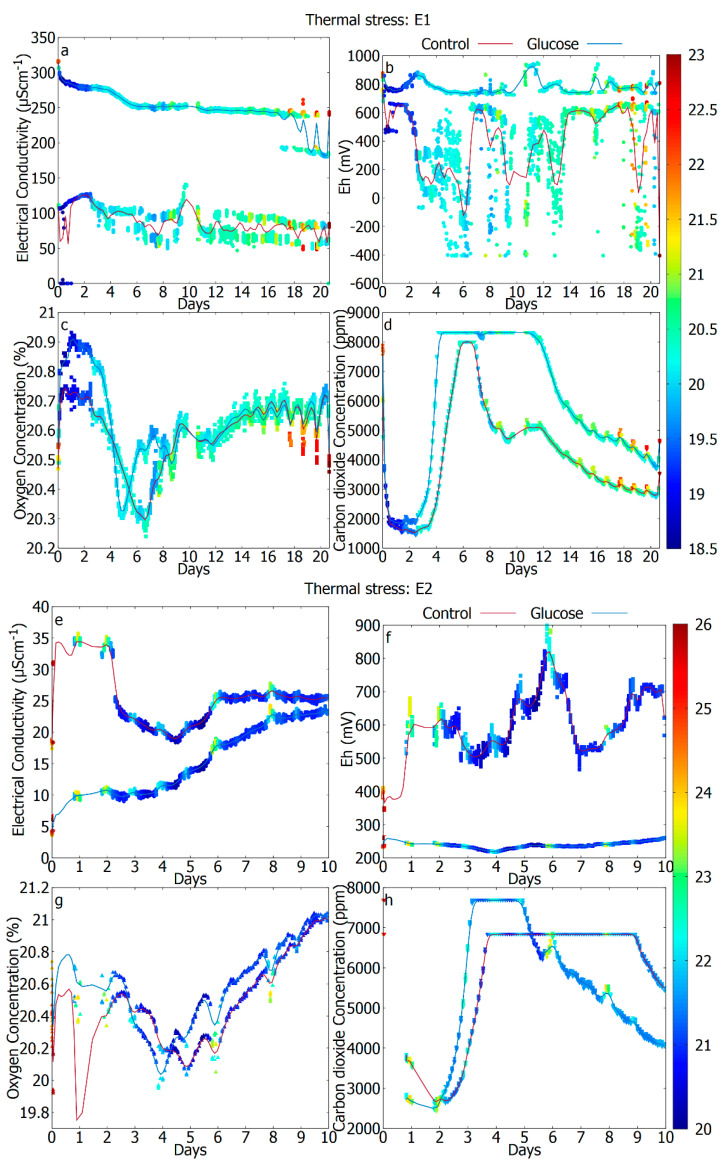
Experiment II. Results are showing an initial dormant period of the metabolism followed by a reactivation for control (red) and glucose (blue) cases of the incubated soil treated under thermal stress. E1: (**a**) electrical conductivity (EC), (**b**) redox potential (Eh), (**c**) oxygen concentration, (**d**) carbon dioxide concentration, E2: (**e**) electrical conductivity (EC), (**f**) redox potential (Eh), (**g**) oxygen concentration, (**h**) carbon dioxide concentration. The color bar represents the air temperature (°C).

**Figure 4 sensors-20-04479-f004:**
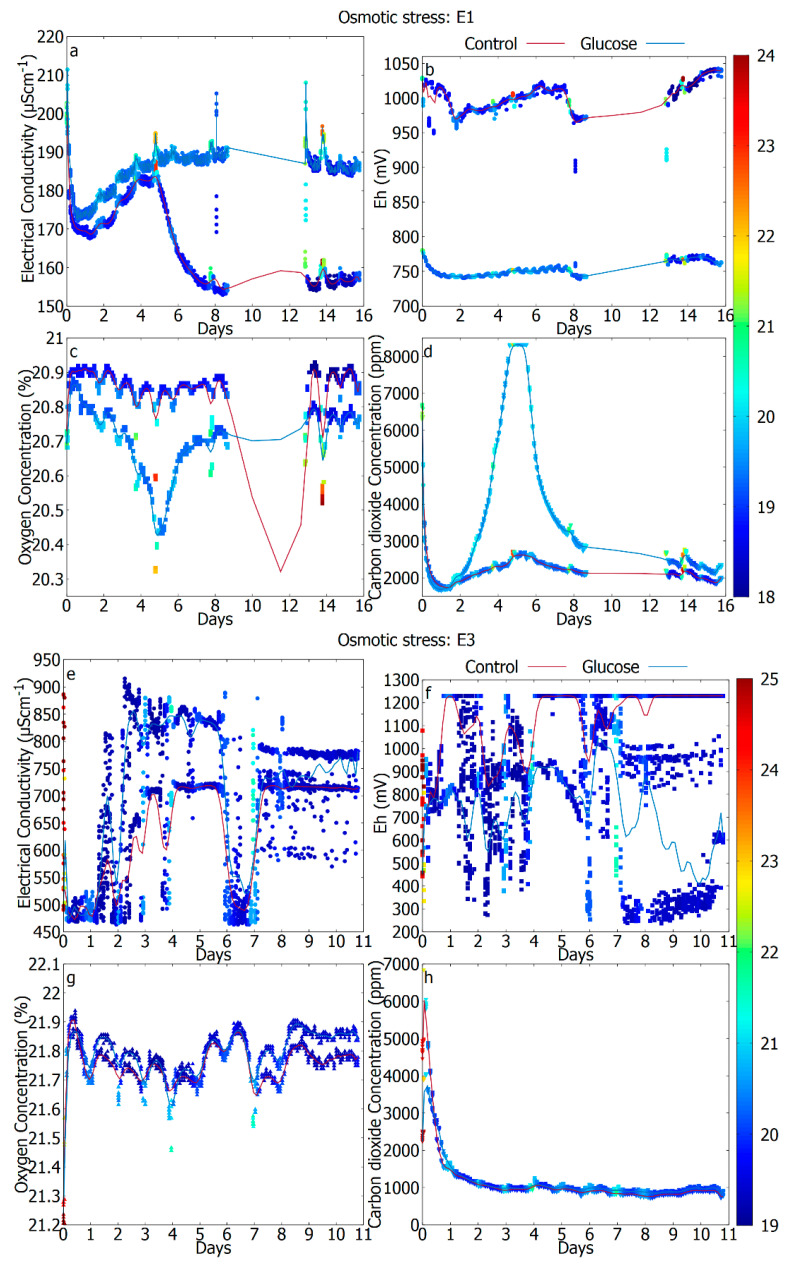
Experiment III. Results are showing an initial dormant period of the metabolism followed by a reactivation for control (red) and glucose (blue) cases of the incubated soil treated under osmotic stress. E1: (**a**) electrical conductivity (EC), (**b**) redox potential (Eh), (**c**) oxygen concentration, (**d**) carbon dioxide concentration, E3: (**e**) electrical conductivity (EC), (**f**) redox potential (Eh), (**g**) oxygen concentration, (**h**) carbon dioxide concentration The data gaps in (**a**–**d**) were because of the data logging problems. The curve at the data gaps is due to the cubic Bezier smoothing of the entire set of data points. We recommend omitting this part for interpretation. The color bar represents the air temperature (°C).

**Table 1 sensors-20-04479-t001:** Technical specifications of the Metabolt sensor suite.

Sensor/Probe	Product Model	Measurement	Measurement Range	Resolution
Waterproof temperature sensor	DS18B20 1-Wire Digital Thermometer	Soil temperature	−55–125 °C	0.25 °C at 10-bit
Two 50 × 30 × 0.5 mm copper sheets 90 mm apart	Atlas Scientific EZO™ Conductivity circuit	Electrical conductivity	0.07–500,000 µS/cm	±2%
3-Pt (Custom-built by Paleo Terra, The Netherlands. Platinum sensing elements at distances 10, 20, and 30 mm, respectively, from the tip allowing for redox measurements at different depths of the experiment samples) redox probes and Ag|AgCl wire with 3M KCl reference probes	Atlas Scientific EZO™ ORP circuit	Redox potential	−1019.9–1019.9 mV	±1 mV
Oxygen sensor	CO2Meter UV flux 25% oxygen sensor module	Air pressure Oxygen concentration	500–1200 mbar 0 to 25 %	1.1 mbar 0.01%
Carbon dioxide sensor	CO2Meter COZIR Ambient 10,000 ppm CO2 + RH/T sensor	Air temperature Relative humidity Carbon dioxide concentration	−25–55 °C 0–95% 0–10,000 ppm	0.08 °C 0.08% 1 ppm

**Table 2 sensors-20-04479-t002:** Experimental conditions performed in this study. Each column denotes a different soil treatment and was run for three experiments: E1, E2, and E3. The experiment conditions with three different treatments and the information about each experiment are also provided.

Condition	(I) Unperturbed	(II) Thermal Stress	(III) Osmotic Stress
Soil pre-treatment	No pre-treatment	Heated at 125 °C for 35.4 h	No pre-treatment
Added solution (control experiment)	40 mL of deionized water	40 mL of deionized water	40 mL of deionized water
Added solution (actual experiment)	40 mL of deionized water + 0.5% glucose (0.2 g)	40 mL of deionized water + 0.5% glucose (0.2 g)	40 mL of deionized water + 0.5% glucose (0.2 g) + 25% CaCl2 (10 g)
Number of experiments (E)	3 (E1, E2, E3)	3 (E1, E2, E3)	3 (E1, E2, E3)
Incubation time (days)	E1: 10 E2: 9 E3: 15	E1: 20 E2: 10 E3: 18	E1: 16 E2: 14 E3: 11
Soil sample collection season	E1: Autumn 2017 E2: Summer 2018 E3: Summer 2018	E1: Autumn 2017 E2: Summer 2018 E3: Summer 2018	E1: Autumn 2017 E2: Summer 2018 E3: Summer 2018
Figure description	E1: Not shown E2: Figure 2a–d E3: Figure 2e–h	E1: Figure 3a–d E2: Figure 3e–h E3: Not shown	E1: Figure 4a–d E2: Not shown E3: Figure 4e–h

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
