# Peer review of "Metabolt: An In-Situ Instrument to Characterize the Metabolic Activity of Microbial Soil Ecosystems Using Electrochemical and Gaseous Signatures"

_sensors, 2020, doi:10.3390/s20164479_

Round 1

Reviewer 1 Report

The authors appear to have addressed most of the earlier criticisms in this new text. I am still not convinced that the ca. 8000 ppm ceiling is due to exchange but rather looks like a limit in the sensor for CO2. This should be double-checked. The range of experiments/soils is still rather limited but maybe sufficient as an example of the capabilities of Metabolt. I still find the system to be quite rudimentary and not all that novel.

Author Response

The authors appear to have addressed most of the earlier criticisms in this new text. I am still not convinced that the ca. 8000 ppm ceiling is due to exchange but rather looks like a limit in the sensor for CO2. This should be double-checked. The range of experiments/soils is still rather limited but maybe sufficient as an example of the capabilities of Metabolt. I still find the system to be quite rudimentary and not all that novel.

Answer: Thank you for the clarification. The 8,000 ppm ceiling is certainly the saturation in the sensor. Once the experiment starts and there is a build-up of CO2, the sensor measurements shows a higher value. But after multiple uses of the sensor, due to offset correction in the initial value, the highest measurable value was about 8,000 ppm instead of the original maximum range of 10,000 ppm for the new sensor.  The behaviour of saturation of the sensor is already provided in lines 231-234, but for clearing this confusion we have also included text in lines 249-251.  

We agree with your opinion about experimenting Metabolt with different soils and temperature conditions to explicitly point out what kind of studies will be best suited for a Metabolt like instrument concept and will certainly do so in the future. Furthermore, as a measurement system to aid future space exploration, Metabolt could serve as a new instrument concept fulfilling the requirements of a simple yet robust for space operations, which has driven us to include the basic measurement principles which have a long history in microbial ecosystem monitoring field that can be adapted as a space instrument.

We are planning on continuing this study and maturing the instrument concept with more experiments such as the ones you suggested, so that we would find the right implementation out of the range of expected applications.  

Reviewer 2 Report

This manuscript focused on the design, instrument calibration, and the performance of the Metabolt instrument to characterize the metabolic activity of microbial soil ecosystems using electrochemical and gaseous signatures. This paper has great novelty and a nice structure. To improve the quality of this paper, some comments are as follows.

  1. The novelty of this article should be clarified. Is there any other similar instrument, and the state-of-art related work should be discussed in comparison with the proposed Metabolt instrument, and its performance.
  2. There are numerical typos throughout the references, the author should check the citation form consistency carefully.
  3. For detection, engineers use various types of sensors. Vision methods are the future of non-contact monitoring for materials and structures. Extensive works have been done on the displacement, crack, and deformation detection using these methods. Please add related works in the introduction to improve the readability (Real-time detection of surface deformation and strain in recycled aggregate concrete-filled steel tubular columns via four-ocular vision; High-accuracy multi-camera reconstruction enhanced by adaptive point cloud correction algorithm).

Author Response

This manuscript focused on the design, instrument calibration, and the performance of the Metabolt instrument to characterize the metabolic activity of microbial soil ecosystems using electrochemical and gaseous signatures. This paper has great novelty and a nice structure. To improve the quality of this paper, some comments are as follows.

1. The novelty of this article should be clarified. Is there any other similar instrument, and the state-of-art related work should be discussed in comparison with the proposed Metabolt instrument, and its performance.

Answer: Thank you for your suggestion. We have discussed the related work in the paragraph in lines 66-91 giving credits to some of the key previous works on various sensing methods used in the Metabolt instrument and how the combination of approaches makes Metabolt novel.

2. There are numerical typos throughout the references, the author should check the citation form consistency carefully.

Answer: Thank you for pointing it out. We found that few references were jumbled up and have corrected those. Now, it should be correctly numbered along with few other fixes.

3. For detection, engineers use various types of sensors. Vision methods are the future of non-contact monitoring for materials and structures. Extensive works have been done on the displacement, crack, and deformation detection using these methods. Please add related works in the introduction to improve the readability (Real-time detection of surface deformation and strain in recycled aggregate concrete-filled steel tubular columns via four-ocular vision; High-accuracy multi-camera reconstruction enhanced by adaptive point cloud correction algorithm).

Answer: Thank you for your suggestion. Of course, non-contact optical measurements would be good and reliable. We did think of few optical methods for studying microbial activity such as measurements of dissolved oxygen in the sample, but at this stage of the instrument, since we use unconsolidated raw soil samples without any separate liquid extraction for measurements (which is exactly the point of the instrument to investigate microbial activity directly on the samples), we think optical sensors will be much noisier and hard to interpret for in-situ measurements. However, few gas sensors we used, for example, the NDIR CO2 sensor is based on optical measurements and are more reliable.

Your suggested methods would however be useful to monitor over time the growth of macroscopic structures with cameras, the possibility to monitor the growth of biofilms, fungi structures and microbial mats or other macroscopic structures on the surface of the soils with small sized cameras and advanced algorithms you kindly quoted. This may be of particular interest for the future application in space greenhouses. We have added a comment in lines 87-91 regarding this.

Round 2

Reviewer 2 Report

accept

This manuscript is a resubmission of an earlier submission. The following is a list of the peer review reports and author responses from that submission.

Round 1

Reviewer 1 Report

In the article construction of a portable soil incubator and examples of its application for soil studies are described. The underlying idea is to reconstruct soil functionality in natural conditions in apparatus which can determine physico-chemical properties of soil sample. In the soil incubator called Metabolt electrical conductivity, redox potential and concentration of gaseous components in headspace can be determined.

Remarks

R. 1. Metabolt function is listed in l.102:

"1. mimic natural growth conditions;" - current Metabolt design represents semi-open thermodynamical system. Vapours and gases can irreversibly leave soil sample. For example, humidity of soil can significantly decrease during experiment. The losses are not supplemented by transport from deeper soil layers or surrounding. As a result, physico-chemical parameters of the soil sample are altered in comparison to the soil in the natural conditions.

Though, the character of semi-open system can be regarded as a feature, not defect. The measurements results can lead to valid conclusions for designed experiments, like the ones described in the article. But for in situ observations a modified, more "open" version of Metabolt should be used. For example, it can be a frame with sensors (a box without bottom) pushed into the soil surface. 

R. 2. Is Meatbolt the first apparatus constructed for microbiologocal activity studies in soil? If not, mention former results in text.

R. 3. l. 207 "... relative humidity (RH) ..." – Is RH determined in soil or in headspace air?

Reviewer 2 Report

General comments

This paper describes the initial testing of an instrument designed to detect microbial activity in soil (and presumably any regolith material). While the instrument is well-described (particular details are in the supplementary information), I feel that the data presented is too scant to give a proper assessment. Only one soil was tested. It is never explained why EC measurements should indicated microbial activity – more on this is required. The Eh values were not always as predicted and again this requires explanation. Apart from the unperturbed soil, the temperature and osmotic stressed soil samples gave predicable results for carbon dioxide but some unexplained results for the other parameters. It is not totally clear what might be a machine response to temperature (or other external influences) and what is due to microbial activity. Throughout, I get the impression that the authors do not have a full grasp of microbial ecology. The allusions to circadian rhythms should be omitted as the observed diurnal variations can be explained by temperature fluctuations. Some data is not presented and it is not clear whether this was to save space or because the results were unacceptable. Really there should be much more basic testing.

Specific comments

L22 Replace “on” with “in”.

L36 Is “Greenhouses” really justified as a keyword; it is only mentioned a couple of times?

L53 Throughout, the emphasis is on bacteria whereas many soils are dominated by fungi. Their role in soil metabolism should be acknowledged.

LL55-57 This statement is inaccurate and misleading. The current application of Metabolt does not mimic natural conditions since the regolith sample has to be removed, sieved and placed within the sample chamber. I would even question the use of the word “in-situ” in the title, though the meaning is more that the instrument can be used in the field as opposed to the laboratory. There are many tools for measuring gaseous exchange between soil and atmosphere which have been around for a long time and many are truly in-situ, e.g. automated soil respiration chambers. Of course, there is always the problem that to measure anything you potentially disturb the system in some way. Similarly, continuous in situ Eh measurements have been applied in many contexts – one recent example is Reiser et al. (2020) who combine it with oxygen uptake. What is perhaps more unique (to my knowledge) is the fact that Metabolt combines these different approaches with electrical conductivity in one system – this fact should be emphasised more.

LL74-84 I think the evidence for circadian rhythms in soil bacteria is speculative or at least very weak (Sartor et al. 2019). It is perhaps understandable that microbes that depend on higher organisms (plant roots, mammalian guts) for a regular nutrient supply may have potential for a circadian response, but general (non-rhizosphere) organisms are not likely to have such an in-built “clock”. Any diurnal changes are most likely a direct response to daily temperature changes (see below).

L113 Only oxygen and carbon dioxide are monitored here (as in many other studies). It would be interesting to know how feasible it would be to have similar gas sensors for the other gases mentioned in L390.

L390 I think CH4 should be added to this list. The inclusion of H2O is strange; I am not sure if changes in H2O could be monitored in any meaningful way. Although it is a product of metabolism, any viable system would need a background of H2O to be sustained.

L134 What is the rationale behind having a soil weight of 250 g? Of course, the weight is to some extent arbitrary but you would get similar results using 100 g or less. Perhaps with <5 g you might encounter random effects. Perhaps the electrode dimensions define the scale of operation? If you add all these weights, it comes to 1.9 kg, not 1.8 kg.

L164 Really, you have used just one Arctic soil, just with three treatments – reword. However, it does appear that it was sampled on two occasions (Table 2). It would be useful to add some more details about the soil such as soil type, depth of collection, organic matter content, moisture content and pH.

LL173-174 Normally, one would use steam sterilization (121 ËšC for 15-20 mins) for soil though any sterilization process alters the soil chemically as well as microbiologically. Since your soil would start off moist, I suspect that the reduction in viable cells would be far in excess of what you cite. In any case, your calculation is not for total viable cells but for bacterial spores, which only form a fraction of the soil microbial population; any vegetative cells would be killed many times over. It also raises the question of what would happen to Metabolt if subjected to this treatment (as would be required for space exploration)?

L185 (Table 2) Since TR1 was collected at a different time from TR2 and TR3, they are not “technical replicates”. It is curious that TR1 was collected after the other two but comes first in the study. It is not explained why some of the data is not presented in the figures.

L190 “200 g” – is this dry weight or fresh weight?

L192 Size of sieve mesh?

L197 Replace “in 40 ml” with “in the 40 ml”.

L216 Does this correction assume a linear degradation?

L223 How was the smoothing done (over what number of data points)?

L229 Do not use “carbon hydrate”, replace with “carbohydrate”.

L231 Ambient (outside) CO2 should be closer to 400 ppm and often is higher indoors.

L234 Clearly, there is an exchange of both gases with the external environment since the chambers are not airtight. Otherwise the system would likely run out of oxygen with glucose addition and go anaerobic. Hence the CO2 also reaches a kind of equilibrium level at ca. 8000 ppm where production equals leakage from the chamber. There is a definite diurnal variation in the data – clearer in O2 than in CO2. While ambient O2 will vary anyway (due to light photosynthesis/dark respiration), the data suggests a more direct response of the soil to temperature. This is very obvious in TR3 for O2 and to a lesser extent for CO2. It would have been useful to have a run without soil as another control.

L242 2b, not 2d. I do find it strange that the Eh for glucose is higher than the control; I would have expected the reverse. In 2f (TR3) it is the reverse. There are similar anomalies in Figure 3 – how is this explained?

L244 Again this response seems to be temperature-controlled. I suspect that this may be a machine response rather than actual changes in the soil.

L247 Use of the word “circadian” implies some internal (biochemical) control, whereas I suggest that the changes are purely in response to daily changes in temperature.

L251 (Figure 2). It is slightly misleading to have the legends for “Control” and Glucose” at the top of the figures – it looks like the left hand side are controls and the right are glucose-treated. I suggest you omit or relocate (it is in LL 251-252 anyway). It is unclear why there are missing data points around 6 days (in a, b and c) but yet there is still a smoothed curve over this period. The fact that the point colour represents the temperature (soil or air?) should be stated.

L256ff. These results are to be expected in a non-sterile system. Sterilized soil will contain many labile nutrients (then enhanced with the glucose). These will enable rapid colonization by contaminants (or survivors, if any). Two to three days at these temperatures is what you might expect for growth to become measurable.

L260 See comment on this above.

L285 (Figure 3) There are issues with the CO2 hitting a ceiling value, which makes interpretation difficult. Also, why are the initial values so high (3000-4000 ppm, not 300-400 ppm)?

L317 (Figure 4) The data here is quite confused. In particular, the electrical measurements in TR3 seem to be very unstable. I am also unclear how you can have meaningful electrical conductivity measurements in a system with 25% CaCl2 added.

LL323-342 This is discussion, not results.

L354 To be frank, I am not clear how the EC and Eh results from the current study can be used to indicate microbial metabolism. There are few clear trends and many unexplained deviations and contradictions.

L362 You cannot distinguish dead (or even dormant) cells with these methods; only the response of living cells.

L398 The Metabolt instrument has only been tested on one soil (treated in three ways). Yes, there is mention of the data from the Mine (but results here were not that clear cut). I would suggest that the instrument requires much more extensive testing with a range of regoliths, and ideally with some under constant temperature to eliminate the confounding effects of temperature.

Reiser et al. (2020) System for quasi-continuous simultaneous measurement of oxygen diffusion rate and redox potential in soil

JOURNAL OF PLANT NUTRITION AND SOIL SCIENCE

Volume: 183

Issue: 3

Pages: 316-326

Sartor et al. Are There Circadian Clocks in Non-Photosynthetic Bacteria?

Biology (Basel). 2019 Jun; 8(2): 41.